# School-based obesity prevention for busy low-income families—Organisational and personal barriers and facilitators to implementation

**Åsa Norman**[1]*, **Gisela Nyberg**[1,2], **Anita Berlin**[3]

**1** Department of Public Health Sciences, Karolinska Institutet, Stockholm, Sweden, **2** The Swedish School of Sport and Health Sciences, Stockholm, Sweden, **3** Department of Neurobiology, Care Sciences and Society, Karolinska Institutet, Huddinge, Sweden

* asa.norman@ki.se

## Abstract

### Background

Little research has targeted multiple-level barriers and facilitators in school-based parental support programmes. This qualitative study aims to describe barriers and facilitators, at organisational and personal levels, that teachers and parents in disadvantaged settings in Sweden perceived as influencing the implementation of the Healthy School Start II (HSS II) intervention.

### Methods

Data collection, analysis and interpretation were guided by the Consolidated Framework for Implementation Research (CFIR). Focus groups and interviews were conducted with 14 parents and ten teachers within the HSS II trial. Data were analysed using qualitative content analysis in a deductive step using the three CFIR domains–inner and outer setting, and personal characteristics–followed by an inductive analysis.

### Results

The theme 'being on the same page–getting burdened teachers and parents to work on common ground' was found. Among teachers, barriers and facilitators were related to the structure of the schoolwork and curriculum, involvement from other staff and school management, the practical school workday, perception of high family needs but low parental interest, insufficient resources in the families, and teacher's personal knowledge, interests, and opinions about health and food. For parents, barriers and facilitators were related to the perceived family needs and resources, parents' health knowledge, consensus about healthy behaviours and ability to cooperate, and school involvement in health issues and the intervention.

**Data Availability Statement:** Data cannot be shared publicly because of ethical reasons, where public availability would compromise participant privacy. Data are available from the research group

'Community Nutrition and Physical Activity', Department of Public Health Sciences, Karolinska Institute (contact via registrator@phs.ki.se) for researchers who meet the criteria for access to confidential data.

**Funding:** GN received funding from Martin Rind Foundation, http://www.martinrind.nu/. GN & ÅN received funding from Sven Jerring Foundation, https://jerringfonden.se/. The funders had no role in study design, data collection and analysis, decision to publish, or preparation of the manuscript.

**Competing interests:** The authors have declared that no competing interests exist.

**Abbreviations:** CFIR, Consolidated Framework for Implementation Research; HSS, A Healthy School Start; MI, Motivational Interviewing; SEP, socio-economic position.

## Conclusion

Interventions should facilitate parents' and teachers' work on common ground, with activities suitable for a stressful and burdensome workday and everyday life. This could be achieved by integrating evidence-based practices within school routines, and including activities that are practicable despite parents' stressful lives, and that increase parental consensus about promoting health. Strategies to increase involvement of parents in families with high needs are necessary. Also, this study suggests an expansion of the CFIR to capture the interface between different micro-level organisations, and account for several delivering/receiving organisations.

## Introduction

Obesity comprises a serious threat to health in a global perspective [1], with large socioeconomic differences in high-income countries [2]. In Sweden, the prevalence of child obesity is at least three times higher in families with low socioeconomic position (SEP) relative to families with high SEP [3]. The need for interventions targeting prevention of overweight and obesity early in life has been widely emphasised internationally [4, 5]. Parents are essential in such interventions as they constitute gatekeepers for their children's healthy dietary and physical activity behaviours, especially in younger years. Interventions including parents have also shown favourable effects in systematic reviews including high-, and upper middle-income countries [6, 7]. However, it is important that interventions are well adapted to the contexts of both deliverers and participating parents [8, 9]. This may be particularly vital in settings with low SEP as disadvantaged families are often difficult to recruit and retain in health interventions [10], and deliverers may experience burdensome workdays, impeding their ability to implement a new health intervention [11]. A recent review and summary of international guidelines and recommendations for child obesity prevention described that school has been identified as an important arena for obesity prevention in Europe, North America and other high- and middle-income countries [12]. The review concluded that, in addition to structural actions for child obesity prevention, school-based obesity prevention that include both teachers and parents are emphasised for pre-school and school-aged children [12]. Little research has investigated how barriers and facilitators on different levels, such as factors at organisational and personal levels, affect school-based parental support programmes, especially regarding parental perspectives. Previous studies investigating organisational and personal factors that influence the implementation of school-based obesity prevention interventions have shown that schools as organisations are ideal for delivering health messages [9]: the personal role of teachers is important [13]. However, time and resources are scarce in schools; the demanding work environment offers little scope for additional health work in the form of projects, and little support from the rest of the school [9, 14, 15]. Clear prioritisation from those in charge of policy and politics is needed [16].

The Healthy School Start (HSS) is an intervention to promote healthy dietary and physical activity behaviours and to prevent child overweight and obesity through school-based parental support. The intervention was developed to suit families with low SEP [17]. The programme has been evaluated in two cluster-randomised wait-list controlled trials, the HSS I in 2010–2011 in areas with medium to low SEP [18], and the HSS II in 2012–2013 in areas with low SEP [19].

This study comprises part of the process evaluation of the HSS II trial, complementing previously published quantitative [19] and qualitative [8] process data from the intervention. This qualitative study reports on barriers and facilitators related to organisational and personal factors influencing implementation of the intervention by using three of the five domains/levels in the Consolidated Framework for Implementation Research (CFIR) [20]: 'inner and outer settings', and 'personal characteristics'. A previous publication reported on factors related to the HSS II intervention itself, and the process of implementing the intervention using the other two CFIR domains 'intervention characteristics' and 'process' [8]. The previous study reporting on the HSS II found that tailoring interventions to the participants' capabilities is important in increasing their engagement [8]. The quantitative process evaluation of the HSS II trial showed good reach and programme fidelity [19].

The above introduction highlights the need to implement early prevention of child obesity through parenting interventions. and to increase knowledge on potential facilitators and pitfalls to the implementation of such interventions. This study aims to describe barriers and facilitators, at both organisational and personal levels, that teachers and parents in disadvantaged settings perceived as influencing the implementation of the HSS II intervention.

## Methods and materials

### Study design

This study employed a qualitative descriptive design which is useful in process evaluations as it permits studying patterns, perceptions, and processes related to an intervention [21].

### Setting

The HSS is a school-based parental support intervention targeting health promotion of children's dietary and physical activity behaviours in the home environment; it is specifically designed for families and schools in disadvantaged areas and conducted in pre-school classes (5-7-year-old children). The programme is based on Social Cognitive Theory [22]. The study protocol has been published [17], and comprises three core components:

1. Health information to parents as a brochure with evidence-based information on children's diet and physical activity developed specifically for the programme based on a literature review [23]. It is written in an easy-to-read format to cater to parents with low proficiency in Swedish, and was pre-tested by parents in the targeted areas.

2. Motivational Interviewing (MI) [24] for parents focusing on their child's specific dietary or physical activity behaviours at home that they wanted to change. The face-to-face session lasted for a maximum of 45 minutes and was attended by one or both parents without the child. A follow-up session focusing on the targeted behaviour (face-to-face or by telephone) was offered three months later. The MI sessions were conducted by skilled MI counsellors who were part of the research team.

3. Classroom activities for the children developed in collaboration with pre-school teachers. Ten 30-minute lessons were delivered by the teachers according to a teacher's manual, which mirrored the themes in the parental brochure, and were accompanied by a workbook that the children were to complete at home together with their parents.

The programme was evaluated during 2012–2013 as a cluster-randomised controlled trial with wait-list, the HSS II trial. The trial included 378 families in 13 schools with 31 pre-school classes (intervention n = 16, control n = 15) in disadvantaged areas in Stockholm, Sweden, where the obesity prevalence is ten times higher than in affluent areas [25]. Significant

intervention effects were found regarding lower intake of unhealthy foods and drinks, and lower BMI in obese children compared to the control group [19]. The quantitative process evaluation of the HSS II trial showed that parents of 146 children participated in the first MI session, of whom 86 also participated in the second, follow-up session. The MI counsellors performed MI with documented quality. Teachers performed 8–10 lessons in each class with an average of 33 minutes per lesson. All 10 home assignments were completed by 12 classes, whereas the remaining 4 classes completed between 1 and 8 of the assignments [19].

## The Consolidated Framework for Implementation Research

In this study the Consolidated Framework for Implementation Research (CFIR) was used as a base for data collection, analysis, and interpretation. Building on 19 different published frameworks, theories, and models for implementation of interventions within the health services, the CFIR [20] consolidates concepts found to influence outcome to a single typology employing consistent terminology. In total, CFIR comprises 39 different constructs spread over five overarching domains: 1. intervention characteristics; 2. outer setting; 3. inner setting; 4. characteristics of individuals; and 5. process. This study focuses on three of the five settings reflecting organisational and personal factors influencing implementation:

- Outer setting–comprising four constructs which refer to patients' needs and resources, peer pressure, external policy, and incentives.

- Inner setting–comprising 12 constructs which refer to structural characteristics of the implementing organisation, networks and communication, culture, implementation climate, and readiness for implementation such as leadership engagement.

- Characteristics of the individuals–comprising five constructs which refer to knowledge and beliefs about the intervention, self-efficacy, individual stage of change and identification with the organisation, and other personal attributes.

   Data collection focusing on barriers and facilitators relevant for the implementation of the HSS programme was guided by the three domains in this study. The three domains were chosen as they were regarded as relevant for the implementation of the HSS programme. However, as the constructs of each domain are very specific, it was not considered meaningful to sort data into the specific constructs; instead, the general meaning of the domain was interpreted as described in the points above. The CFIR informed data analysis in a deductive analytical step and was also used in the final interpretation of the results. The deductive analysis made it clear that the domains inner and outer context corresponded to different 'organisations' for the parents and teachers. In order to make meaningful deductive categories of data, domains were applied differently for parents and teachers. Thus, parents' inner setting corresponded to the family whereas parents' outer setting corresponded to the school, or society at large. For teachers, inner setting corresponded to the school whereas outer setting corresponded to the families, or society at large (see Fig 1).

## Participants

Regarding parents, we initially aimed for a purposeful sample with maximum variation [21] of sex of the parent and child, parental country of birth, participating schools, the family's degree of participation, and target behaviour in the MI sessions of the HSS II trial. A sample of 45 parents were drawn from the HSS intervention group (185 families). However, approximately 75% of the parents declined participation due to lack of time and a second recruitment phase was initiated. Now all parents who had participated in a least one MI session (104) were

| Informants | Inner setting | Outer setting |
|---|---|---|
| Parents | Family | School and society at large |
| Teachers | School | Family and society at large |

**Fig 1. Reference to how CFIR domains were used in the deductive stage of data analysis.**

invited, which resulted in a recruitment of eight to eleven parents in each of the four focus groups, of whom a total of 14 ultimately participated, representing nine of the 16 intervention school classes. Four to seven parents per focus group cancelled the appointment without advance notice or simply did not attend sessions (Table 1).

**Table 1. Characteristics of participating parents and teachers.**

| Parents n = 14 | |
|---|---|
| Mothers/fathers (n) | 9/5 |
| Education maximum 12 years (n) | 8 |
| Age (mean years (range)) | 38 (30–49) |
| Parent of girl/boy/twins (n) | 8/5/1 |
| School classes represented (n) | 9 |
| Born in the Nordic region [a] (n) | 6 |
| Mean (range) years in Sweden if born outside the Nordic region | 20 (10–40) |
| Countries represented outside the Nordic region | Afghanistan India Iraq Korea Lebanon Somalia Turkey |
| Target behaviours in MI sessions | Overweight/food Activity/eating together Sleep Parental influence on child Activity/food Vegetables Activity Variation of food No focus Food |
| **Teachers (n = 10)** | |
| Women | 10 |
| Age (mean years (range)) | 51 (29–63) |
| Education/profession (primary school/pre-school teacher) (n) | 4/6 |
| Work experience in pre-school class (mean years (range)) | 9 (1.5–25) |

[a] Including Sweden, Norway, Denmark, Finland, and Iceland

All 21 teachers in the intervention group of the HSS II trial were invited to participate; ten teachers agreed, representing seven of the 16 intervention school classes (Table 1).

This study used indicators of socioeconomic position (SEP) on both group and individual level [26, 27]. Group level indicator of SEP was represented by area of residence where all three residential areas in this study were characterised by low employment, low level of education and blocks of flats [28]. Individual level indicator of SEP was represented by self-reported parental level of education (Table 1).

## Data collection

Focus group methodology was chosen as data were generated through participant interaction, which makes it suitable for describing experiences [29, 30]. Separate interview guides were constructed for teachers and parents and consisted of open-ended questions based on the CFIR [20], where each question was linked to a specific CFIR domain. Guides were pilot-tested on one parent and one teacher resulting in minor changes to increase understanding. Examples of questions to teachers were: What was it like to work with the programme? Describe pros and cons about working with the programme. How was the programme received by the rest of the school? How do you perceive your own capability to work with programme? How do you perceive the parents' capability to work with programme? How do you perceive the importance of working with these issues at school? In what way was school leadership involved in the programme? What capacity does the school have to carry out this type of programme?

Examples of questions to parents were: What was it like to work with the programme? Describe pros and cons about working with the programme. How did you perceive your capability to work with the programme? How do you perceive the relevance of the programme in relation to your family? How did you perceive the dedication to the programme on the part of the school? How do you perceive the communication and information regarding the programme?

As focus groups should be formed according to a common characteristic among participants [29], parents and teachers were placed in separate focus groups where teachers shared the role as intervention deliverers and parents were placed in groups according to the number of MI sessions they had attended. In total, four focus groups were conducted with parents where two focus groups included parents who had participated in one MI session (four parents in each group) and two focus groups included parents who had participated in two MI sessions (with four and two parents, respectively). Two focus groups were conducted with teachers (four teachers in each group) and semi-structured interviews were done with two individual teachers to compensate for focus group drop-out. All focus groups were moderated by ÅN and included one assistant. During the focus groups and interviews, continuous summaries and an end summary were made by the moderator and commented upon by the participants to ensure credibility of data [31, 32]. All focus group were audio recorded and carried out in October-December 2013.

## Ethical considerations

All participants consented in writing. Ethical approval was obtained from the Regional Ethical Review Board in Stockholm, Sweden (2012/877-31/5).

## Data analysis

Data analysis was undertaken using qualitative content analysis [33] in two phases, first a deductive phase followed by an inductive phase, according to Elo & Kyngäs [34]. The analysis was conducted by ÅN and peer-reviewed by AB to ensure trustworthiness [31, 32]. During the

analyses a reflexivity journal was kept by ÅN and AB to ensure confirmability of the results [32]. Steps undertaken in the analysis were: first ÅN listened to and transcribed the audio recordings, after which ÅN read the transcripts several times. Second, relevant data corresponding to barriers and facilitators to implementation of the HSS programme were deductively sorted into the three domains of CFIR: 'Inner setting', 'Outer setting', and 'Personal characteristics'[20]. Third, data that had been sorted in each of the three CFIR domains were analysed inductively; data for each domain was analysed separately. Data indicating barriers and facilitators were identified and coded using the open coding technique in accordance with Elo & Kyngäs [34]. Fourth, patterns were searched for and identified among codes, and codes were merged into subcategories. Fifth, patterns were searched for and identified within subcategories, which were merged into categories. In the sixth and final step, subthemes covering the inductive categories and an overarching theme covering all data were identified. Categories, subthemes, and the overarching theme were discussed among all authors until consensus was reached. In the analysis, data from teachers and parents were initially treated separately, but merged at the time of identifying subthemes, in the sixth step of analysis. Thus, inductive categories representing both parents and teachers were merged and included under the three identified subthemes. Quotes in the text include ellipses, modifications and explanations within square brackets to increase comprehensibility. Participants in each focus group were assigned a number (1–4), to ensure anonymity. When quoted in the text, participants are labelled teacher (T), or parent (P) with an assigned participant number together with the number of the focus group (FG) or interview (I). All data were collected and transcribed in Swedish and translated into English during the fifth step of analysis.

## Findings

The findings are summarised in Fig 2. Below we describe first the latent overarching theme and its relation to the manifest categories, and then the deductive categories with corresponding inductive categories comprising barriers and facilitators.

| Theme | Being on the same page – getting burdened teachers and parents to work on common ground | | | | | |
|---|---|---|---|---|---|---|
| Subthemes | Fighting for resources and conditions to focus on health | | Frustrating disagreement on needs and involvement | | Role of own interest, knowledge and convictions | |
| Inductive categories Barriers (B) Facilitators (F) | **School work structure and task** **B:** 1.other school values, 2.dense pre-school class curriculum 3.lack of – a. communication with staff, b. planning time, 4. many ongoing projects **F:**1.pre-school class curriculum – a.flexible, b.involves 'health' 2.'health' a school value | **Parental consensus on healthy behaviours and cooperation in the family** **B:**lack of agreement on healthy behaviour between parents **F:**parental agreement on healthy behaviour/ responsibilities | **Teachers' view of family needs** **B:** parental ignorance of unhealthy habits and parenting in families with the most need **F:** parental engagement in healthy habits and parenting | **School involvement** **B:** lack of – a.dietary products at school, b.consensus on health between school/home, c.school engagement **F:** 1.staff promotes health at school, 2.healthy school meals, 3.consensus on health between school/home | **Knowledge of the subject** **B:** lack of knowledge of specific issues **F:** 1.good subject knowledge, 2.extensive work experience | **Knowledge of the subject** **B:**1.high level of prior knowledge about health, 2.lack of knowledge on parenting of the subject |
| | **Involvement from the rest of the school** **B:** 1.lack of leadership engagement, 2.project invisible in the school | **Family needs** **F:** identified needs of the child | **Teachers' view of family resources** **B:** 1.lack of - a.money, b.time, c.language skills, d.knowledge, 2. culture 3.experiences of hardship 4. migration status **F:** knowledge, money | | **Interest in the subject** **F:** 1.interest based on children's needs, 2.personal interest | |
| | **The school workday** **B:** staff turnover, sick-leave etc. | **Resources in the family** **B:** lack of time for joint family activity/meals | **Teachers' view of parental attitude towards the school** **B:** lack of -1.interest in child school work, 2.communicat. with school | | **Personal opinions about the subject** **B:**1.opposite opinion about diet, 2.negative view on HSS activity | |
| Deductive sorting | Inner setting | Inner setting | Outer setting | Outer setting | Personal Characteristics | Personal Characteristics |
| Informants | Teachers | Parents | Teachers | Parents | Teachers | Parents |

(left axis: Process of analysis — Phase 2 / Phase 1)

**Fig 2. Description of analysis and findings.** Phase 1: deductive sorting into CFIR domains, phase 2: inductive identification of categories, subthemes and theme.

## Overarching theme: Being on the same page—Getting burdened teachers and parents to work on common ground

Data from parents and teachers revealed that barriers and facilitators within the CFIR domains personal characteristics, inner, and outer setting circled around teachers and parents being on the same page in terms of working with the HSS intervention specifically, and health generally. Getting teachers and parents to work on common ground regarding health seemed challenged by teachers' burdensome everyday work and parents' demanding life situation.

Teachers found, on the one hand, that their task and the curriculum facilitated work with health, but on the other hand that the cumbersome structure, and everyday work at school often made it difficult to prioritise, or get attention from the school leadership or other staff for their specific health work. The latter resulted in the teachers fighting for resources and latitude to work with the health activities. The teachers also identified that the families with the most urgent need to work with health behaviours did not involve themselves in the HSS programme, and teachers found it frustratingly difficult to work on common ground.

Parents found that an identified need in the family and consensus between parents facilitated family health action. However, parents also struggled for resources and conditions to focus on health activities amidst an arduous family life with difficulties finding time, or agreeing on health issues within the family. In addition, the parents found a lack of consensus with the school concerning what health messages to convey to children, that the school was not deeply enough involved in the health work, and identified unaddressed needs for change within the school itself, mostly regarding food. Thus, parents identified actions on the part of the school that counteracted working on common ground. In addition, mismatches between parental knowledge, and teachers' interests, knowledge, personal convictions, and opinions regarding health acted as barriers to working on common ground to influence the children's health.

## Inner setting—Barriers and/or facilitators among teachers

**School work structure and task.** The structure of the schoolwork and tasks for the school in general and specifically for the pre-school class acted as both barriers and facilitators to working with the HSS programme.

On the one hand, the pre-school class curriculum was described as a facilitator for carrying out the HSS programme as it is flexible and has a clearer focus on health than the curriculum for the higher grades.

> [O]lder pupils have to have certain hours of Swedish or maths and then it can be hard figuring out what subjects these [the HSS activities] are and where they fit. But in pre-school class, it's just perfect.
>
> (I 2)

On the other hand, teachers described how the specific work structure in pre-school class also acted as a barrier, as most of the working hours are spent with the children, with very little joint planning time. Teachers imagined that the programme might spread more easily if it were conducted in higher grades where teachers have more time to plan and communicate with each other.

> T1. I think it would be more established [in higher grades]. Because then teachers and teaching assistants have more time to talk to each other, daily. More than what you have in

pre-school class, where it's basically once a week that everyone can talk together about what's going on and plan.

(FG 1)

In addition, the pre-school curriculum was described as very dense, with many subjects and activities among which the teachers themselves were to set priorities; this acted as both a facilitator and a barrier to implementing the programme, depending on the teachers' choices. The teachers described how each pre-school class had its own activities based on the specific teacher's choice, where some cared for many of the other subjects whereas others chose to skip some of the other subjects to get more time for HSS.

T3. For us [teachers in pre-school class] it is a bit up to you. How you organise [classwork] and how much you want to work with [different subjects]. T1. There are so many other things we have to do, we have language groups, we have natural science and technology, and we have the maths workshop and so on. T3. Yeah.

(FG 2)

Furthermore, the fact that school is an arena for many different projects acts as a barrier to sustainability of HSS. The teachers described how new projects are constantly being started only to disappear after a short time. This leaves teachers with a feeling that nothing is ever finished properly or leaves any traces in the schools.

T3. Yeah, it's a bit tough [with new projects all the time]. And then then they may not turn out very well. They're not completed. And then something new comes along and then something new... T4. We're used to new things coming up all the time [laughing]. T3. [And] then disappearing

(FG 2)

Specific school values acted as both barrier and facilitator: values focusing on e.g. health or environment acted as a facilitator for the HSS, whereas other school values acted as a barrier.

T1. [If] they had done a big project in the entire school, I don't think many [staff] would have seized upon it. Because what this school is interested in and considers important is language [...] T3. Our school has worked a lot on this before also and we call ourselves a 'health school'.

(FG 1)

**Involvement from the rest of the school.** Involvement from staff and the visibility of the HSS in the school constituted both barriers and facilitators for conducting the HSS. A half-hearted involvement from the head of the school constituted a barrier: teachers described how principals would generally only ask once or twice how the programme work was going. Most teachers found this disappointing but were used to this situation: being obliged to do something without communication or feedback. Some teachers asked themselves if the principal joined the programme because it would 'look good'. Some teachers called for the principal to show more interest as a sign of appreciation, whereas others didn't see how the principal could be more involved.

T2. Yes, but if it was the school management that has wanted this at the beginning then [. . .] I don't understand what their objective was [. . .]. To be honest, is it supposed to look good that we're participating in this study? T1. Yeah, [the principal came] at the start and asked, when I said that we had had the healthy school start, 'Oh, yeah, how's it going?', something like that but nothing more. T2. No, she never asks me. T1. Only briefly.

(FG 2)

The HSS was described as invisible to the rest of the school, which constituted a barrier and teachers wished for more time to inform others in school about the programme.

So the others [school staff] don't have any specific experience of it [. . .] But there are so many things going on in a school all the time so. . . it was a bit invisible.

(I 1)

**The school workday.** The teachers described how day-to-day school events comprised barriers for the HSS. The dynamics of everyday school life made it difficult to foresee what each day would be like; sick-leave and acute situations, as well as continuous, time-consuming reorganisation and staff turnover resulted in marginalisation of projects like HSS.

Back then [at the time of the HSS] many of us hadn't yet worked together. And for the first time there were three classes instead of two so there was a lot to talk about, how should we organise the work, how should classes be formed and what should we do.

(I 1)

## Inner setting—Barriers and/or facilitators among parents

**Parental consensus on healthy behaviours and cooperation in the family.** Agreement, cooperation, or division of responsibility between parents regarding diet and physical activity in the family constituted both barriers and facilitators to how the HSS was implemented. In families where domestic work was strictly divided and the parent with an interest in diet was also in charge of cooking, that same parent was often the one involved in HSS and implemented part of the programme in the family as she/he saw fit. Responsibility for programme then fell to only one parent, which seemed to facilitate implementation. Parental division of domestic work, if combined with disagreement or conflicts about diet, acted as a barrier.

P3. Because at our house, I'm the one who finds it very important to eat healthily and [the other parent] doesn't find it as important. And we both do the same amount of cooking, so (laughter) it gets. . .. (P4. A conflict there.) Yes! And he's more OK with unhealthy snacks between meals than I am.

(FG 3)

**Family needs.** Parents described how a perceived need in the family, in the form of a high or low food intake or low activity level of the child, comprised a facilitator to working with the programme.

P3. One of our four children doesn't move much and eats a lot and he has a completely different body than the others too. So, it's good that this [project] was for him.

(FG 3)

**Resources in the family.**   Parents described preconditions in the form of time constraints and stress as barriers to maintaining joint physical activity and mealtimes in line with the recommendations from the HSS material.

P2. Sure, the [project material] says that you're supposed to sit together the whole family and make the meal calm and relaxed, but the fact is that the way our lives are there's stress all the time.

(FG 5)

## Outer setting—Barriers and/or facilitators among teachers

**Teachers' views of family needs.**   An identified need among the parents, or perceived lack of need, facilitated or acted as a barrier to implementation of the programme, according to the teachers. Many teachers perceived a substantial need in the families related to child weight, health behaviours, and parenting. They described that many children in class were overweight, that many families appeared to consume an unhealthy diet, and that parents seemed to have difficulties setting limits or that they infused sedentary and obesogenic behaviours by providing sweets or allowing the child a lot of screen time. However, many teachers described a lack of parental identification of needs as a barrier: the parents who engaged in the programme were those who already had healthy habits, whereas families with high needs did not engage.

T1. But these children get to direct and choose what they want themselves, the parents have no power over their children [and. . .] they don't have very healthy dietary habits, I suppose. T3. No! No, and activity, no activity habits either, none at all. [. . .] T2. [On an excursion] one child had five thick hot dogs. That's what he had for lunch, only hot dogs. (T1. Yeah) No vegetables, and a soda. [. . .] T4. It was all wrong, the slim kids participated and the active kids with active parents, they participated, but the kids who really needed it, we couldn't get them involved.

(FG 2)

**Teachers' views of family resources.**   Resources in terms of family finances, language skills, education, and time, as well as parental background and migration status, comprised barriers and facilitators. The teachers described how some of the families could not afford to send the children to organised activities, buy them unusual fruits to taste or pay for after-school care. Consequently, these children go home and spend the afternoons in front of a screen.

T3. I think there's one [child] in three classes who plays football [in a club]. Because that costs money, and activities are very, very expensive. If they were free then everyone would join. Of course, because they play football during every break.

(FG 2)

Some teachers experienced how low proficiency in Swedish caused several parents to misunderstand or not be able to complete the HSS activities, whereas others described that language did not pose a barrier. In addition, teachers described how several of the immigrant families were newly arrived, which posed barriers as teachers did not have time to help everyone in the manner needed. Teachers discussed how the parents' cultural background affected health habits, attitudes towards school and parenthood and imagined that previous experiences, for example famine, might result in a perception that everything edible is good.

> T3. Well, they read and don't understand what this is all about, many of them. T1. They have problems understanding the spoken language and even more problems reading.
>
> (FG 2)

The teachers perceived the parents' knowledge of diet and physical activity as variable, acting as both facilitator and barrier: some parents were well-informed whereas others lacked knowledge and were unreceptive towards information.

In addition, the teachers described the parents' demanding everyday life as a barrier, where long or inconvenient working hours or having many children left little time and energy to help the children with schoolwork.

> T1. That's a difficulty too for parents who work different hours and may not see each other very much either (T3. No, exactly) and even less their children. They don't have a chance to give their children that part [engagement in health activities].
>
> (FG 1)

**Teachers' views of parental attitude towards the school.**   The teachers described general difficulties in engaging parents in schoolwork as a barrier for implementing the HSS. Teachers found it hard to get parents to come to school, read information, or be involved in the children's schoolwork.

> T2. It's a struggle to get parents to come here [to school] and to capture their interest.
>
> (FG 2)

## Outer setting—Barriers and/or facilitators among parents

**School involvement.**   Parents' perceptions of staff engagement in the programme and health practices at school constituted barriers and facilitators for the HSS. Some parents stressed expressed the importance of home and school sending similar messages regarding diet and physical activity; a lack of vegetables and fruits and an ignorance of principles for healthy food at school made it difficult to get the child to eat healthily at home. Other parents were content with the school's food practices and content.

> P4. Even if you have very clear rules at home, if the school doesn't really adhere to the rules you have at home, e.g. to eat according to the Healthy Eating Plate, things we [at home] talk a lot about how important they are, if the schools don't comply then it fails.
>
> (FG 3)

Parents experienced school engagement in the programme differently. Strong school engagement was a facilitator, whereas some parents were frustrated by unclear engagement from the pre-school class teachers, and lack of engagement from the school management and canteen. All parents desired better cooperation between school and home regarding projects like the HSS.

P2. I think it's a disaster, if a school joins a project like this then I think the management and canteen and everyone [at the school] should comply with the same [message. Otherwise] they might just as well not join. But on the other hand, the commitment our teachers showed was really good.'

(FG 5)

### Personal characteristics—Barriers and/or facilitators among teachers

**Knowledge about the subject.** The teachers' perceptions of their own level of knowledge acted as a barrier and facilitator to how HSS was carried out. Some teachers wished for more specific information on how to respond to the children, whereas others felt that educated teachers had enough knowledge.

T1. Yes, the stuff about minerals and energy was difficult to explain to the children. T4. Yeah, at least for me it's about not having enough. . . I don't know enough myself.

(FG 1)

**Interest in the subject.** The teachers' personal interest in diet and physical activity, both to meet the children's needs and for their own sake, acted as a facilitator for conducting HSS.

For me personally, diet and exercise mean a lot. And if I can communicate that to my students and their families, that's just an advantage.

(I 2)

**Personal opinions.** The teachers' own personal standpoints in relation to parts of the programme material acted as a barrier to conducting the HSS. Some teachers did not feel convinced about the dietary advice, and others disliked certain activities and simply chose to skip them.

T2. Well, I did do what [the material] said and now I've been sitting here saying it was all so great, but I don't know if I always think the Healthy Eating Plate model suits everyone.

(FG 2)

### Personal characteristics—Barriers and/or facilitators among parents

**Knowledge about the subject.** Parents' knowledge about food comprised a barrier. Some wished for more knowledge on interaction with the child regarding the home assignments or healthy habits, whereas others felt that the HSS did not contribute to anything new, but appreciated the programme nonetheless.

P2. [In the MI session] we talked about food, but I already knew everything since I'm involved in that myself quite a lot [. . .] P1. But if we're expected to do it [workbook] exclusively at home, then I think there should be a bit more information material for me as a parent.

(FG 4)

## Discussion

This study describes barriers and facilitators on organisational and personal levels that influence the implementation of a school-based parental support programme to promote health and prevent child overweight and obesity. Overall, the study shows that it is essential to support and facilitate for teachers and parents to work on common ground regarding children's health, and to ensure that the intervention does not increase the burden on participants and deliverers, but rather facilitates their work and everyday lives. Thus, school-based health interventions to support health in the home environment need to be structured in such a way that it is easy for parents and teachers to work together, and difficult to deviate from the intervention structure, or drop out of the programme.

### Facilitating teachers' health work—Creating standard practice and involving the rest of the school

For teachers, working with a health intervention such as the HSS programme needs to be easy. Indeed, the structured material and activities have been described as easy to use by teachers [8]. The teachers in this study perceived the pre-school class curriculum as flexible, with its focus on health comprising a facilitator: all in all a promising starting point. However, teachers also described barriers: the dense curriculum, lack of planning time, lack of support from principals and other staff, and how projects come and go without feedback or sustainability. These barriers to working with health according to the HSS programme are in line with those identified in previous studies on school-based health promotion in Sweden [11, 15] and elsewhere [9, 14, 16]. To further facilitate and create sustainability for health promotion efforts, the HSS programme should become standard practice, not something that teachers can choose themselves and/or that is perceived as extra work. Describing the HSS activities in greater detail in the curriculum or other guiding documents for pedagogical work in pre-school classes would integrate the work as routine and reduce the likelihood that teachers view it as extra work or as yet another programme that comes and goes.

The CFIR emphasises the importance of leadership engagement and of involving the rest of the organisation for successful implementation of an intervention [20]. This kind of dedication was requested by both teachers and parents in the study. Including the health work as part of the curriculum would require that school leadership be involved to ensure sustainability. Furthermore, adding similar health work to the curricula in higher primary school grades would get more school staff involved in the work. In this intervention study the MI sessions were conducted by research staff. Future MI sessions will be delivered by school nurses as part of evidence-based routine to work with health promotion. This will further increase the involvement of the rest of the school in the health promotion work.

At present, no evidence-based health promoting programmes are being used in Swedish schools, or in school health care. An integration of the HSS programme into standard school practice could result in all teachers working with evidence-based health promotion, doing the

same tasks in the same way. The work would not be dependent on any individual person, which would enhance the sustainability of the structured health activities.

### Facilitating parental health work—Motivating, and supporting cooperation with burdened parents

Making the health work of the HSS programme part of the school curriculum and standard practice for school health care will facilitate getting parents and teachers to work together, as health activities would then be part of both the children's mandatory schoolwork and the health care system for school children.

It is well known that families with a disadvantaged position in society are difficult to engage and retain in health interventions [10]. There may be several reasons for this, but the stressful everyday life has been emphasised in several studies [35, 36]. Teachers in this study voiced frustration when they realised that the parents of children with the greatest need for health promotion were less likely to engage in the HSS programme. Teachers also described their difficulties getting parents in these disadvantaged areas to come to meetings at school or engage in the children's schoolwork. Thus, there is an evident need for specific activities focusing on motivating parents to engage in the health promoting work delivered at school. Previous studies have shown that parental knowledgeability about nutrition [37] and insights regarding long-term health concerns [38] can facilitate motivating parents to work with health promotion at home. We believe that a motivation strategy should be used to engage parents to promote their children's healthy behaviour at school start. Possible strategies include giving parents brief feedback on the child's health status and presenting strategies for lifestyle changes. This could be integrated into the health assessment done routinely by the school nurse or physician during the child's the pre-school class year in Sweden. Another strategy might be to offer parents a short test (e.g. for the risk of diabetes type-2) and feedback on their own health status. Within the family, disagreement between parents on what constituted healthy behaviour was a barrier. Parents also called for more advice on how to interact with their child concerning food and physical activity. This is in line with other studies; qualitative studies have described how parents stress the importance of learning parenting skills related to setting limits, role modelling, and parenting as a team around healthy eating [35, 38]. A recent study also commented that including fathers and father figures is important in order to motivate a healthy lifestyle within the family [39]. Thus, to further increase parental commitment and engagement, future implementations of the HSS programme should include a greater emphasis on parenting strategies and facilitate cooperation between parents in the family.

### Adapting the CFIR to cater to multi-layered analysis

The CFIR is an implementation framework based on the idea of complex interaction between an intervention and several contextual domains that influence successful implementation of the intervention [20]. The original description of the CFIR states that the model and its constructs do not depict 'interrelationships, specific ecological levels, or specific hypotheses' [19]. However, CFIR has been used widely within studies focusing on implementation of health interventions [40] in which an ecological underpinning is often essential to understanding the intervention context and target group. The HSS intervention is based on the social ecological model [41] and includes two different micro-level systems: the school and the family. This study used the CFIR to guide both data collection and deductive qualitative analysis. Deductive qualitative analysis is used to test or extend a theory or model [42], in this case the CFIR itself. The deductive analysis highlighted difficulties in applying the CFIR to data comprising two microsystems (i.e. school and families). Firstly, data that included the interface between school

and family were difficult to sort into any one domain. Here we refer to data where parents discuss e.g. the involvement and responsibility of the school and vice versa. The social ecological model accounts for the interface between different microsystems through the mesosystem [41]. We find this a potentially important part of implementation which is difficult to grasp when using the CFIR to understand data. A second difficulty arose when we attempted to fit the data to the CFIR domains. As described in the methods section, our analysis revealed that the inner and outer settings of deliverers (teachers) and recipients (parents) corresponded to different organisations. The CFIR framework assumes one single organisation as the delivering organisation. This is problematic as several complex interventions involve more than one organisation. Also, a model that considers only the delivering organisation further exacerbates the problem, as it makes the receiving organisation invisible. In the case of this process evaluation, that would lead to omission of data from parents, and loss of important information on barriers and facilitators to implementing the HSS intervention. Based on this discussion we suggest some modification and expansion of the CFIR as a theoretical framework to better include the interface between different micro-level organisations, and to account for several delivering and/or receiving organisations. We believe this would make the CFIR more useful in guiding implementation of complex interventions.

## Strengths and limitations

This study contributes to the field of implementation science in several ways. An important strength of the study is the use of implementation theory in the form of CFIR complemented with an inductive phase of analysis. The use of CFIR throughout the data collection, analysis and interpretation makes the findings transferable to other contexts, which is of great importance in implementation research [40]. In addition, the inductive analysis phase of this process evaluation fills a gap, providing more detailed and context-based information about barriers and facilitators in the implementation of obesity prevention interventions. Such information has been called for to further tailor interventions to contextual needs and preconditions [9]. Furthermore, the contribution to theory development of the widely used implementation framework CFIR constitutes an important strength of the study. In addition, illustrative quotes, intersubjective agreement in the analysis process, and the audit trail described in the methods section vouch for the study's trustworthiness [32]. The study was not conducted strictly according to the suggested use of the CFIR for post-implementation investigation, which comprises a limitation [20, 40]. Furthermore, recruiting parents in disadvantaged settings is known to be difficult [10]. That was also the case in this study, which limits the transferability of findings.

## Conclusion

The findings indicate a need for interventions to facilitate parents' and teachers' work on common ground, with activities suitable for a stressful and burdensome workday and everyday life. This could be achieved by integrating novel evidence-based practices into school routines to facilitate school staff's work with health promotion without unduly increasing the burden. In addition, programme activities should be made suitable for parents' stressful lives, adapted to a variety of cultural norms, and include activities to increase parental consensus to promote health. Furthermore, strategies to increase involvement and motivation of parents in families with high needs and low resources are needed. In addition, this study suggests an expansion of the CFIR as a theoretical framework to better include the interface between different micro-level organisations, and to account for several delivering and/or receiving organisations.

## Implications

School-based health work like the HSS should:

- Be integrated into the school curriculum to facilitate teachers working uniformly and possibly evidence-based and to facilitate sustainability

- Be integrated within general school routines, and supported by school management and staff to ensure sustainability

- Include activities to stimulate constructive parental discussion about family health habits

- Include motivational strategies in order to reach target groups with elevated needs by

  - Providing more information and hands-on guidance to parents on positive parenting to promote children's healthy behaviours

  - Giving clear instructions to the school nurse on how to motivate parents e.g. during health assessments

  - Finding ways to give dialogue-based feedback to parents on family health habits

## Acknowledgments

We thank all parents and teachers participating in the study. We also thank the Sven Jerring Foundation and the Martin Rind Foundation for funding the study.

## Author Contributions

**Conceptualization:** Åsa Norman, Gisela Nyberg, Anita Berlin.

**Data curation:** Åsa Norman.

**Formal analysis:** Åsa Norman, Anita Berlin.

**Funding acquisition:** Gisela Nyberg.

**Methodology:** Anita Berlin.

**Supervision:** Anita Berlin.

**Writing – original draft:** Åsa Norman.

**Writing – review & editing:** Gisela Nyberg, Anita Berlin.

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
