## [Decision Letter · Decision Letter 0]

30 Aug 2019

PONE-D-19-15288

School-based health-promotion for busy low-income families – organisational and personal barriers and facilitators to implementation

PLOS ONE

Dear Ms Norman,

Thank you for submitting your manuscript to PLOS ONE. After careful consideration, we feel that it has merit but does not fully meet PLOS ONE’s publication criteria as it currently stands. Therefore, we invite you to submit a revised version of the manuscript that addresses the points raised during the review process.

ACADEMIC EDITOR: You will see that the peer reviewers have all suggested fairly minor revisions to improve this manuscript. These largely relate to typographical or grammatical errors, although there are some helpful suggestions around locating the intervention and the study in an international context and commenting on the efforts taken to enhance the credibility of the findings. Good luck with making these changes.

We would appreciate receiving your revised manuscript by Oct 14 2019 11:59PM. To enhance the reproducibility of your results, we recommend that if applicable you deposit your laboratory protocols in protocols.io, where a protocol can be assigned its own identifier (DOI) such that it can be cited independently in the future. For instructions see: http://journals.plos.org/plosone/s/submission-guidelines#loc-laboratory-protocols

We look forward to receiving your revised manuscript.

Kind regards,

Shelina Visram, PhD, MPH, BA

Academic Editor

PLOS ONE

Journal Requirements:

2. Please ensure you have included the registration number for the clinical trial referenced in the manuscript.

Reviewers' comments:

Reviewer's Responses to Questions

**Comments to the Author**

1. Is the manuscript technically sound, and do the data support the conclusions?

Reviewer #1: Yes

Reviewer #2: Yes

Reviewer #3: Yes

2. Has the statistical analysis been performed appropriately and rigorously? 

Reviewer #1: Yes

Reviewer #2: N/A

Reviewer #3: Yes

3. Have the authors made all data underlying the findings in their manuscript fully available?

Reviewer #1: Yes

Reviewer #2: Yes

Reviewer #3: Yes

4. Is the manuscript presented in an intelligible fashion and written in standard English?

Reviewer #1: Yes

Reviewer #2: Yes

Reviewer #3: Yes

5. Review Comments to the Author

Reviewer #1: Thank you for giving me the opportunity to review this paper. I find it to be an interesting topic. The focus of this paper is a good illustration of the on-going effort to combat obesity as a health issue among school children, and also the example of how schools and parents can play their roles in such effort. This paper also provide a thick description of the intervention program that was conducted in Sweden, thus highlighting the importance of developing an intervention program that suit to the context of the study. The findings give the insights into the ways of dealing with challenges in implementing such intervention. In general, this paper could be considered for publication. Nonetheless, the author could enhance the paper’s rigour by considering these suggestions:

1. In the introduction section, it was a good attempt made by the authors to focus on school-based obesity prevention intervention as the main focus of the study. However the title of the paper does not clearly reflect this . Therefore, it is suggested that the title of the paper should specifically mention “school-based obesity prevention” as to align it with the focus of the study.

2. Also in the introduction part, the authors have succeeded in highlighting obesity as a health issue in Sweden. The discussion would be more comprehensive if other international contexts are included. For instance, how the same issue is being addressed within the other contexts (such as the European context) should also be discussed.

3. While the authors managed to highlight the issue, the organization of the paper would be more structured if the authors include the statement of problem that reflects the background of the problem, followed by research objectives and research questions.

4.With regards to methodology, the authors have provided extensive details of how data had been analyzed through content analysis and how trustworthiness had been achieved. Nevertheless, additional relevant literature is needed to support the process of validation of qualitative data. For instance, participant verification or other strategies like inter-rater agreement in validating qualitative data need to be considered. The following articles can be found useful:

• Hayes, A. F., & Krippendorff, K. (2007). Answering the call for a standard reliability measure for coding data. Communication methods and measures, 1(1), 77-89.

• Morse, J. M. (2015). Critical analysis of strategies for determining rigor in qualitative inquiry. Qualitative health research, 25(9), 1212-1222.

Reviewer #2: The study reported in the paper is a school-based program to prevent obesity among children attending schools located in a disadvantaged areas of Sweden. Overall, the authors have fully described the evaluation aimed at identifying the barriers and facilitators to the implementation of the school-based program. The authors have provided qualitative data highlighting perspectives of parents involved in the project and teachers who implemented the intervention.

There are a few issues which the authors need to address. Details are provided below:

Abstract

1. There is need to indicate that the study was conducted in Sweden so that readers get this important information from reading the abstract.

2. It is important for the authors to confirm that the program being evaluated is the Healthy School Start 11 Trial earlier on, perhaps in the background so that readers understand this. The authors need to clarify that the program being referred to in the Methods section is the same as that mentioned in the background.

Introduction

This is well written and clear.

Methods

1. I suggest that the authors present the research process in past tense since the study has been conducted.

2. The typographical errors on line 164 should be corrected.

Findings

This is well written and presented.

Discussion

1. Authors need to correct typographical errors on lines 513, 564.

Reviewer #3: The study presents the results of the original research and the statistics performed are appropriate for a qualitative research. Conclusions are presented appropriately and are supported by the data.

The article is of good quality and some items need to be adjusted:

- Authors should name the intervention variables (eg, a health promotion intervention focusing on overweight and obesity).

- Put in the summary the type of study (qualitative).

- Authors should include in the abstract how the data analysis was performed.

- The purpose at the end of the introduction is different from the one presented in the abstract, so authors should standardize the abstract.

6. PLOS authors have the option to publish the peer review history of their article (what does this mean?). If published, this will include your full peer review and any attached files.

Reviewer #1: No

Reviewer #2: No

Reviewer #3: Yes: Mateus Dias Antunes

---

## [Author Response · Author response to Decision Letter 0]

19 Sep 2019

Response to Reviewers

Thank you for the opportunity to revise this manuscript. We truly appreciate the helpful comments from the reviewers.

Reviewer #1: 

Thank you for giving me the opportunity to review this paper. I find it to be an interesting topic. The focus of this paper is a good illustration of the on-going effort to combat obesity as a health issue among school children, and also the example of how schools and parents can play their roles in such effort. This paper also provide a thick description of the intervention program that was conducted in Sweden, thus highlighting the importance of developing an intervention program that suit to the context of the study. The findings give the insights into the ways of dealing with challenges in implementing such intervention. In general, this paper could be considered for publication. Nonetheless, the author could enhance the paper’s rigour by considering these suggestions:

1. In the introduction section, it was a good attempt made by the authors to focus on school-based obesity prevention intervention as the main focus of the study. However, the title of the paper does not clearly reflect this. Therefore, it is suggested that the title of the paper should specifically mention “school-based obesity prevention” as to align it with the focus of the study.

• The title has been changed to clarify this issue. The title is now: “School-based obesity prevention for busy low-income families – organisational and personal barriers and facilitators to implementation”.

The short title has been changed and is now: “Multiple-level barriers and facilitators to school-based parental support for obesity prevention”

2. Also in the introduction part, the authors have succeeded in highlighting obesity as a health issue in Sweden. The discussion would be more comprehensive if other international contexts are included. For instance, how the same issue is being addressed within the other contexts (such as the European context) should also be discussed.

• A broad description of guidelines in other high-income countries has been added. Also, clarifications have been added to studies that reflect an international context rather than only Sweden. Page 4, lines 61, 64-65, 69-74.

3. While the authors managed to highlight the issue, the organization of the paper would be more structured if the authors include the statement of problem that reflects the background of the problem, followed by research objectives and research questions.

• We agree with the comment and we have included a problem statement (page 5, lines 101-103) to further clarify the background. 

4.With regards to methodology, the authors have provided extensive details of how data had been analyzed through content analysis and how trustworthiness had been achieved. Nevertheless, additional relevant literature is needed to support the process of validation of qualitative data. For instance, participant verification or other strategies like inter-rater agreement in validating qualitative data need to be considered. The following articles can be found useful:

Hayes, A. F., & Krippendorff, K. (2007). Answering the call for a standard reliability measure for coding data. Communication methods and measures, 1(1), 77-89.

Morse, J. M. (2015). Critical analysis of strategies for determining rigor in qualitative inquiry. Qualitative health research, 25(9), 1212-1222.

• A warm thank you for the suggested articles. We have read them with great interest. Also, we agree that references should be added to support the process of gaining trustworthiness to the results of the study. For this purpose, we have chosen to use Lincoln & Guba together with Morse to support our process as we see that both articles follow the same line, where Morse takes the ideas from Lincoln & Guba further. We realise that Morse suggests that concepts should be named ‘validity’ ‘reliability’, etc. Still, we have chosen to continue to use the traditional concepts ‘trustworthiness’, ‘credibility’ etc. as suggested by Lincoln and Guba as we find these suitable for our study. Additional clarifications of the measures taken to ensure trustworthiness of the study are added on lines 225-227, 236-237, 607-608.

Reviewer #2:

The study reported in the paper is a school-based program to prevent obesity among children attending schools located in a disadvantaged areas of Sweden. Overall, the authors have fully described the evaluation aimed at identifying the barriers and facilitators to the implementation of the school-based program. The authors have provided qualitative data highlighting perspectives of parents involved in the project and teachers who implemented the intervention.

There are a few issues which the authors need to address. Details are provided below:

Abstract

1. There is need to indicate that the study was conducted in Sweden so that readers get this important information from reading the abstract.

• The information has now been added on line 26.

2. It is important for the authors to confirm that the program being evaluated is the Healthy School Start 11 Trial earlier on, perhaps in the background so that readers understand this. The authors need to clarify that the program being referred to in the Methods section is the same as that mentioned in the background. 

• This has now been clarified in the background on lines 96, 98, 100, 105. Also, clarifications have been added to the methods section on lines 134, 139, 179, 190.

Introduction

This is well written and clear.

Methods

1. I suggest that the authors present the research process in past tense since the study has been conducted.

• The text in the methods section has been changed to past tense, except for the description of the HSS programme itself, and the CFIR framework. These are both material and framework which are not specific for the how this study was carried out, but serve as base for the study and we therefore believe present tense is more appropriate for these descriptions. 

2. The typographical errors on line 164 should be corrected.

• This has now been corrected, line 180.

Findings

This is well written and presented.

Discussion

1. Authors need to correct typographical errors on lines 513, 564.

• This has now been corrected, line 531, 582.

Reviewer #3: 

The study presents the results of the original research and the statistics performed are appropriate for a qualitative research. Conclusions are presented appropriately and are supported by the data.

The article is of good quality and some items need to be adjusted:

- Authors should name the intervention variables (eg, a health promotion intervention focusing on overweight and obesity).

• The aim of the Healthy School Start intervention itself is to promote children’s healthy dietary and physical activity behaviours in the home environment. This has now been clarified on line 112-113. 

- Put in the summary the type of study (qualitative).

• A description of the study as a “qualitative study” is now included on line 24 in the abstract. 

- Authors should include in the abstract how the data analysis was performed.

• Data analysis in this study were performed using qualitative content analysis in two steps. This is included in the abstract on lines 33-34.

- The purpose at the end of the introduction is different from the one presented in the abstract, so authors should standardize the abstract.

• The aim in the abstract has now been changed.

---

## [Decision Letter · Decision Letter 1]

16 Oct 2019

School-based obesity prevention for busy low-income families – organisational and personal barriers and facilitators to implementation

PONE-D-19-15288R1

Dear Dr. Norman,

We are pleased to inform you that your manuscript has been judged scientifically suitable for publication and will be formally accepted for publication once it complies with all outstanding technical requirements.

With kind regards,

Shelina Visram, PhD, MPH, BA

Academic Editor

PLOS ONE

Additional Editor Comments (optional):

Reviewers' comments:

Reviewer's Responses to Questions

**Comments to the Author**

1. If the authors have adequately addressed your comments raised in a previous round of review and you feel that this manuscript is now acceptable for publication, you may indicate that here to bypass the “Comments to the Author” section, enter your conflict of interest statement in the “Confidential to Editor” section, and submit your "Accept" recommendation.

Reviewer #1: All comments have been addressed

Reviewer #3: All comments have been addressed

2. Is the manuscript technically sound, and do the data support the conclusions?

Reviewer #1: Yes

Reviewer #3: Yes

3. Has the statistical analysis been performed appropriately and rigorously? 

Reviewer #1: Yes

Reviewer #3: Yes

4. Have the authors made all data underlying the findings in their manuscript fully available?

Reviewer #1: Yes

Reviewer #3: Yes

5. Is the manuscript presented in an intelligible fashion and written in standard English?

Reviewer #1: Yes

Reviewer #3: Yes

6. Review Comments to the Author

Reviewer #1: In would like to thank the authors for their efforts in refining the manuscript based on all the comments/suggestions given. I wish the authors all the best for the publication of this manuscript. I look forward to read the final version.

Reviewer #3: The authors made all necessary corrections. This is a great article that presents a relevant topic for health promotion.

7. PLOS authors have the option to publish the peer review history of their article (what does this mean?). If published, this will include your full peer review and any attached files.

Reviewer #1: No

Reviewer #3: Yes: Mateus Dias Antunes

---

## [Editor Report · Acceptance letter]

23 Oct 2019

PONE-D-19-15288R1 

School-based obesity prevention for busy low-income families – organisational and personal barriers and facilitators to implementation 

Dear Dr. Norman:

I am pleased to inform you that your manuscript has been deemed suitable for publication in PLOS ONE. Congratulations! Your manuscript is now with our production department. 

With kind regards,

on behalf of

Dr. Shelina Visram 

Academic Editor

PLOS ONE